# Evaluation of Cortical Bone Microdamage and Primary Stability of Orthodontic Miniscrew Using a Human Bone Analogue

**DOI:** 10.3390/ma14081825

**Published:** 2021-04-07

**Authors:** Chutimont Teekavanich, Masayoshi Uezono, Kazuo Takakuda, Takeshi Ogasawara, Paiboon Techalertpaisarn, Keiji Moriyama

**Affiliations:** 1Department of Maxillofacial Orthognathics, Graduate School of Medical and Dental Sciences, Tokyo Medical and Dental University, 1-5-45, Yushima, Bunkyo-ku, Tokyo 113-8510, Japan; praewteekavanich@gmail.com (C.T.); uezomort@tmd.ac.jp (M.U.); tak2mort@gmail.com (T.O.); 2Department of Orthodontics, Faculty of Dentistry, Chulalongkorn University, 34 Henri Dunant Road, Pathumwan, Bangkok 10330, Thailand; paiboon.t@chula.ac.th; 3Institute of Biomaterials and Bioengineering, Tokyo Medical and Dental University, 2-3-10 Kanda-Surugadai, Chiyoda-ku, Tokyo 101-0062, Japan; tkkdmech@tmd.ac.jp

**Keywords:** orthodontic miniscrew, microdamage, primary stability, pilot hole, synthetic cortical bone

## Abstract

Orthodontic miniscrews have gained popularity; however, they have some drawbacks, including screw loosening that results from bone resorption caused by excess microdamage created during screw insertion. Pilot hole preparation through the cortical bone is considered beneficial to avoid such microdamage, while an overly large pilot hole impairs primary stability. Hence, we used a human bone analogue to evaluate the microdamage and primary stability to estimate the optimal pilot hole size that would minimize the screw loosening risk. Ti6Al4V orthodontic miniscrews and 1.0-mm-thick synthetic cortical bone pieces were prepared. Various compressive loads were applied in indentation tests to test pieces’ surfaces, and the microdamaged areas were confirmed as stress-whitening zones. Screw insertion tests were performed in which a miniscrew was inserted into the test pieces’ pilot hole with a diameter of 0.7–1.2 mm in 0.1-mm intervals, and the stress-whitening area was measured. The insertion and removal torque were also measured to evaluate primary stability. The stress-whitening areas of the 1.0–1.2 mm pilot hole diameter groups were significantly smaller than those of the other groups (*p* < 0.05), whereas the 0.9 and 1.0 mm pilot hole diameter groups showed higher primary stability than other groups. In conclusion, the bone analogue could be utilized to evaluate microdamage in cortical bones and the primary stability of miniscrews.

## 1. Introduction

Orthodontic miniscrews are commonly used in clinics as temporary anchorage devices due to their simple surgical procedure of insertion and performance in absolute anchorage for teeth traction [1]. However, they still have a lower success rate compared to dental implants for prosthodontics [2,3,4].

Recently, the excess microdamage of the cortical bone associated with miniscrew insertion has been reported to play a major role in screw loosening [5]. Microdamage is a permanent deformation of the bone microstructure caused by compressive loads [6]. Two types of microdamage are reported: ‘microcracks,’ consisting of linear defects typically 100 µm in length [7], and ‘diffuse damage,’ comprising a cluster of microcracks that are too small to be distinguished from one another [8]. The accumulation of such microdamage is considered to trigger serial adverse events such as local ischemia, bone resorption, and loosening of the miniscrew [9].

Orthodontic miniscrews are more commonly inserted without pilot holes because the self-drilling procedure is simpler than self-tapping procedure [6], despite greater microdamage occurring with the former procedure [6,9,10]. To avoid excess microdamage, pilot hole preparation through the cortical bone is recommended [6,9,10] as it reduces compressive load to the cortical bone during screw insertion. However, an overly large pilot hole impairs primary stability. Lack of primary stability also results in miniscrew loosening with bone resorption caused by mechanically induced inflammation [11]. Hence, in order to improve the success rate of orthodontic miniscrews, it is important to find the optimum pilot hole diameter that reduces microdamage while providing primary stability. 

There are several articles on the optimum pilot hole diameter for the miniscrew [12,13]. However, in these papers, the diameter was determined with characteristics such as the Periotest value or the insertion torque, which are related to the screw’s stability and not related to the minimization of risk factors related to screw loosening (e.g., magnitude of microdamage). Furthermore, reproducibility in the experiments and extrapolation to clinics might be questionable because rodent or swine models were utilized [12,13,14].

On the other hand, the human bone analogue enabled us to prepare uniform test pieces and was considered ideal for the quantitative evaluation of bone damage. Thus, the objective of this study was to utilize a human bone analogue to evaluate cortical bone microdamage around the miniscrew and to evaluate the primary stability of the miniscrew to estimate an optimal pilot hole size that could minimize the risk of screw loosening.

## 2. Materials and Methods

### 2.1. Specimens

No ethics approval was required because no animal experiments or human studies were involved in this research. Forty-eight 14 mm × 14 mm × 1 mm synthetic cortical bone pieces (#3401-07; Sawbones, Vashon, WA, USA) made of glass fiber-reinforced epoxy resin were prepared as human bone analogues. Eighteen test pieces were used for indentation tests, and the remaining test pieces were used for miniscrew insertion tests. The physical and mechanical properties of the synthetic bone are listed in Table 1. The center point of each test piece was marked with a pencil. Thirty Ti6Al4V miniscrews (Jeil Medical Corporation, Seoul, Korea) that were 1.3 mm in diameter and 6 mm in length were used in miniscrew insertion tests (Figure 1).

### 2.2. Indentation Tests

Indentation tests were performed in which defined compressive loads were applied to the test pieces of synthetic bone. A universal testing machine (AG-X; Shimadzu, Kyoto, Japan) was used with a spherical indenter of 5.0-mm-diameter. Forces of 500 to 1000 N in 100-N steps (six experimental groups, three test pieces per group) were loaded on the center of the synthetic bone surface with a crosshead speed of 1 mm/min.

Following the tests, the tested pieces were embedded in plaster of Paris stained with black ink and cut into two pieces using a low-speed cutter (IsoMet, Buehler, IL, USA) to reveal the cross-sectional surface in the center. The cross-sectional surface was imaged using a light microscope (LV100N POL; Nikon Instruments Inc., Melville, NY, USA) at 20× magnification and imaging software (NIS-Elements; Nikon Instruments Inc., Melville, NY, USA) under controlled light room conditions.

Each image was processed using the image analysis program (ImageJ version 1.51r; National Institutes of Health, Bethesda, MD, USA) to evaluate the microdamage under the applied load. The region of interest was set at 2.0 mm around the center of the indentation. The original images were converted to grayscale images and then binarized with thresholds of brightness ≥ 20 and particle size ≥ 1000 to quantify the microdamaged area (Figure 2A).

To investigate the microdamage of the synthetic bone piece, scanning electron microscopy (SEM; S-4500; Hitachi High-Tech, Tokyo, Japan) observations were also performed. The tested piece loaded with 1000 N was washed with an ultrasonic cleaner with distilled water for 10 s. After Au coating, the cross-sectional surface was observed with a tube voltage of 15.0 kV.

### 2.3. Miniscrew Insertion Tests

The synthetic bone test pieces with a pilot hole were prepared in which the hole diameter ranged from 0.7 to 1.2 mm in increments of 0.1-mm (six experimental groups, *n* = 5 per group). All holes were inspected with light microscopy to confirm the accuracy of the hole diameter. A miniscrew was then manually inserted into the test piece hole and removed with a dial-measuring torque screwdriver (FTD10CN-S; Tohnichi, Tokyo, Japan) to evaluate the primary stability. At the screw insertion, the final gap between the test piece surface and the bottom of the screw head was regulated to be 1.0 mm.

After removal of the screw, the microdamaged areas were quantified in a similar way as the indentation tests. The cross-sectional surface was imaged at 40× magnification. The region of interest for the insertion test was set to 2.0 mm around the center of the hole. The microdamaged area was extracted with a brightness threshold of ≥40 and a particle size of ≥1000 (Figure 2B).

### 2.4. Statistical Analysis

The data from the indentation tests were analyzed using Spearman’s rank correlation coefficient. The data from the miniscrew insertion tests were analyzed using the pairwise Wilcoxon rank sum test adjusted with the Hochberg method. All *p*-values < 0.05 were considered statistically significant. All statistical analyses were carried out using “R” software (version 4.0.2, http://www.r-project.org/ (accessed on 5 July 2020)).

## 3. Results

### 3.1. Indentation Tests

Although no macroscopic cracks were found in any of the test pieces, optical microscopic observations revealed stress-whitened zones approximately 300 µm below the loaded portion for all loading conditions. The intensity of whitening increased as the loading force increased (Figure 3A).

The SEM images showed intact rod-shaped glass fibers embedded in the resin matrix at the area distant from the whitened zone (Figure 3B, left and middle). In the stress-whitened zone, the damaged fibers and cracked matrix with interface failures between them were demonstrated. (Figure 3B, right).

The relationship between the loading force and the area of the stress-whitened zone exhibited a sigmoid curve with a significant increase in the range from 600 to 800 N (Figure 3C). The calculated Spearman’s rank correlation coefficient was 0.95, indicating a high positive relationship between the loading force and the stress-whitened zone (*p* < 0.001). 

### 3.2. Miniscrew Insertion Tests

Fractures of the screw and macroscopic cracks of the tested pieces were not observed in any of the samples. At pilot holes with diameters 0.7 and 0.8 mm, synthetic bone chips came out from the screw hole as the tapping proceeded. In particular, more synthetic bone chips were found with insertion in a pilot hole with a diameter of 0.7 mm.

During optical microscopic observations, thread dents associated with miniscrew insertion were observed in the synthetic cortical bone around the screw hole. The lower margin of the synthetic bone around the screw hole protruded. A stress-whitening zone was observed in the synthetic bone between the dents of the screw thread. These phenomena were clear in the 0.7–0.9 mm diameter group, but unclear in the remaining diameter groups (Figure 4A).

In regard to the relationship between pilot hole diameter and the area of stress-whitening, the area of the stress-whitening zone decreased with the increase in pilot hole diameter, except for in the 0.7 mm pilot hole group. The 0.7 mm pilot hole group showed a smaller area (non-significant) than the 0.8 mm pilot hole group (Figure 4B(a)). The pilot holes with 0.7–0.9 mm diameters showed significantly (*p* < 0.05) larger areas than the other diameter groups.

Insertion torque results revealed significant differences among all groups (*p* < 0.05). Pilot holes with diameters of 0.7 and 0.8 mm had a lower insertion torque than the 0.9 mm pilot hole. In the group with a diameter larger than 0.9 mm, the insertion torque decreased with the increase in pilot hole diameter (Figure 4B(b)).

In the results of removal torque, pilot holes with 0.9 and 1.0 mm diameters showed approximately the same values, which were higher than those of other groups. Except for this point, the removal torque showed the same tendency as the insertion torque. (Figure 4B(c)). In every pilot hole diameter, the removal torque values were lower than the insertion torque values.

## 4. Discussion

During attempts to evaluate cortical bone microdamage around the orthodontic miniscrew and primary stability of the miniscrew, two distinct approaches have been employed. Many studies have attempted animal experiments [6,7,8,9,10,11,12,13,14], whereas several others carried out dry bench tests [15,16]. Each research method involves inherent limitations in mimicking the clinical situation; however, ethical aspects arise as serious issues in the former approach. Russell and Burch claimed the philosophy of the 3Rs: Refinement, reduction, and replacement, regarding the use of animals in scientific experiments [17]. According to such principles, researchers have to be concerned with reducing animal experimentation. Furthermore, since synthetic bones providing similar mechanical properties of human bones are available nowadays, the dry bench test appears to be a more favorable approach. 

Thus, the objective of this study was to evaluate cortical bone microdamage around the miniscrew and primary stability of the miniscrew to estimate the optimal pilot hole size that could minimize the risk of screw loosening by using a human bone analogue. Noticing the fact that the microdamage and the primary stability are the dominant factors in screw loosening, the strategy minimizes bone microdamage while providing primary stability. Here, the microdamage is a permanent deformation of the bone microstructure [6], and the corresponding words in the terminology of mechanics, or the terms applicable to artificial materials, is plastic deformation. The antonyms of these words are reversible deformation and elastic deformation, respectively. Hence, our optimization strategy is the minimization of plastic deformation without the reduction of primary stability in the dry bench test with the use of a human bone analogue.

As a human bone analogue, we used synthetic cortical bone made from glass fiber reinforced polymer (GFRP). When GFRP is subjected to a load that exceeds the yield strength, local congregated crazes and debonding between the fiber and polymer are induced. The outcome of such plastic deformations is the increase in the refractive index of GFRP, which may be observed as stress whitening [18]. In our indentation tests, cross-sectional images of the synthetic bone demonstrated a stress-whitened area, which spread with an increase in the applied load. SEM image of the stress-whitened area revealed damaged fibers and cracked matrix with interface failures between them, i.e., the evidence of the plastic deformation. Considering that the compressive yield strength and elastic modulus of the synthetic cortical bone are the same as those of the human cortical bone [19], it was concluded that the stress-whitening zone in the synthetic cortical bone corresponds to the microdamaged portion in human cortical bone. Therefore, the magnitude of the bone microdamage could be evaluated by measuring the area of the stress-whitening zone induced in the synthetic cortical bone.

The primary stability represents the resistance capacity against the forced displacement under the applied load at the time just after insertion. If the forced displacement is small, the stability of the screw is said to be high. The dominant factors that affect the primary stability are considered to be the area of screw–bone contact and the magnitude of compressive stresses at the screw-bone interface [20], since the grip strength of the surrounding bone onto the inserted miniscrew depends on them. However, these factors are difficult to measure experimentally, so the insertion torque or removal torque was utilized instead [21,22,23]. In the case of a dry bench test with a bone analogue, such usage of the torque can be justified by the following arguments based on elementary mechanics. Firstly, the above-mentioned screw-bone contact and compressive stress generate the frictional force, according to Coulomb’s law [24], as:friction force = μ × compressive stress,
in which μ is the coefficient of friction and the compressive stress is created by elastic deformation in the bone analogue. If the screw is made to rotate, the friction generates the torque as:torque = friction force × screw’s radius,
and the total sum of such torque over the entire surface of the screw constitutes the torque needed to rotate the screw. Here, the insertion torque involves the torque required for the material cutting, whereas the removal torque does not; hence, the removal torque of the screw is directly related to the magnitude of compressive stresses at the screw-bone interface. Therefore, the degree of the primary stability could be evaluated by the magnitude of the removal torque of the screw inserted in the bone analogue.

In the screw insertion tests, the 1.0–1.2 mm diameter pilot hole groups showed almost no stress-whitening zone, which implies that the deformation induced in the bone analogue was mainly elastic, and almost no plastic deformation was present. Among these groups, the elastic deformation became smaller as the pilot hole diameter increased, so the removal torque decreased, and the primary stability also decreased. Thus, 1.0 mm was the optimum diameter among them. On the other hand, the 0.7–0.9 mm diameter pilot hole groups showed significant areas of the stress-whitening zone, i.e., the plastic deformation was predominant. On these groups, it should be noted that the smaller pilot holes did not ensure larger removal torques. This phenomenon might be due to the fact that the plastic deformation was not recoverable and, hence, compressive stresses at the screw-bone interface decreased as the plastically deformed area increased. Thus, the optimum among them was a 0.9 mm diameter pilot hole group with the smallest stress-whitening zone and the highest removal torque. Therefore, comparing these two optimums of two gatherings of groups, the optimum among all groups was 1.0 mm diameter group.

As a consequence, from the evaluation of the corresponding microdamage and primary stability in the human bone analogue, we estimated that the optimum pilot hole diameter was 1.0 mm for miniscrews of 1.3 mm diameter. In the case of such diameter pilot holes, the screw-bone compressive stress during screw insertion was slightly below the yield stress of the cortical bone, and the elastic deformation was maximized. Hence, the plastic deformation of the cortical bone, that is, the microdamage, was minimized, and the removal torque, that is, the primary stability, was maximized. This result is consistent with a previous investigation. Namely, a study utilizing the rat tibia model reported that the optimum pilot hole diameter is approximately 69%–77% of the screw’s outer diameter [12], whereas our results were 76.9% of the outer diameter.

Supplementally, a small stress-whitened area, observed in the 0.7 mm diameter pilot hole group, should be examined. A possible cause was the effect of the self-tapping flute of the miniscrew. Additional micro-CT observation of the screw head revealed that the scoop angle of the cutting edge was approximately zero at the 0.7 mm diameter part, whereas it was negative at the 0.8 and 0.9 mm diameter parts (Appendix A, Figure A1). Hence, in the case of the 0.7 mm diameter pilot hole group, the generation of the cutting chip and the deformation of the materials took place simultaneously during screw insertion, which might have resulted in a decrease in the plastically deformed area of the bone analogue. On the other hand, this mechanism created the thickest plastically deformed region in the 0.8 mm diameter pilot hole group. This region might constitute the structure supporting the compressive stresses from the surrounding elastic region, resulting in the relatively smaller insertion and removal torque observed in the 0.7 or 0.9 mm diameter pilot hole group.

The potential limitations of this study come from the differences between living tissues and artificial materials. Plastic deformations in synthetic cortical bones cannot distinguish linear cracks or diffuse damage in natural bones, although each microdamaged bone would be repaired or resorbed in different ways [25]. Hence, further investigations through clinical experience are required to determine the optimum insertion technique for orthodontic miniscrews.

## 5. Conclusions

Synthetic cortical bone can be utilized to evaluate cortical bone microdamage around the orthodontic miniscrew and primary stability of the miniscrew. The estimated pilot hole diameter that could minimize the risk of screw loosening was 1.0 mm for a 1.3 mm diameter miniscrew.

## Figures and Tables

**Figure 1 materials-14-01825-f001:**
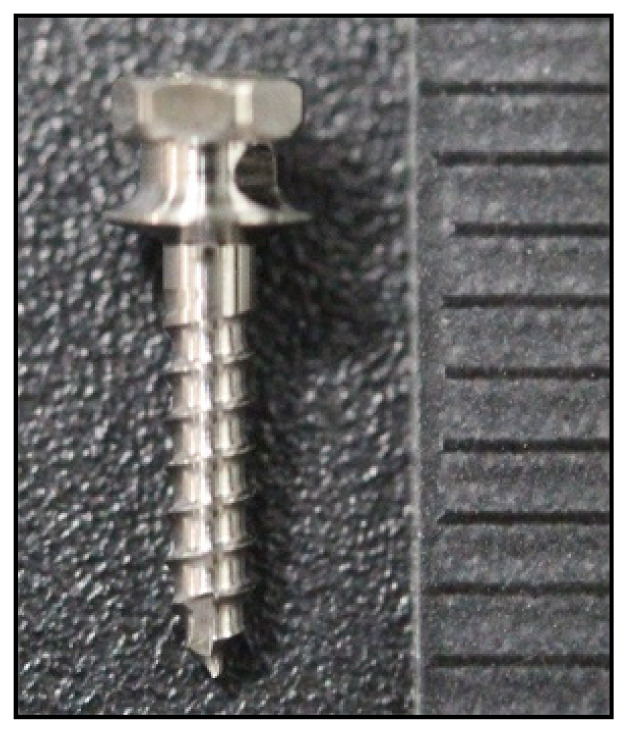
An image of Ti6Al4V miniscrew. Thirty miniscrews (Jeil Medical Corporation, Seoul, South Korea) that were 1.3 mm in diameter and 6 mm in length were used in miniscrew insertion tests. 1 scale = 1 mm.

**Figure 2 materials-14-01825-f002:**
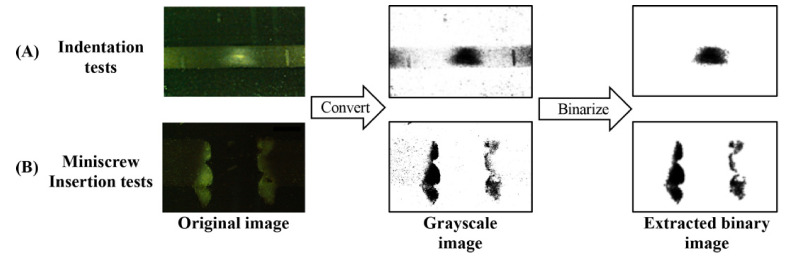
The procedure to extract stress-whitened areas of the synthetic bone using Image J software. (**A**) indentation tests, (**B**) miniscrew insertion tests.

**Figure 3 materials-14-01825-f003:**
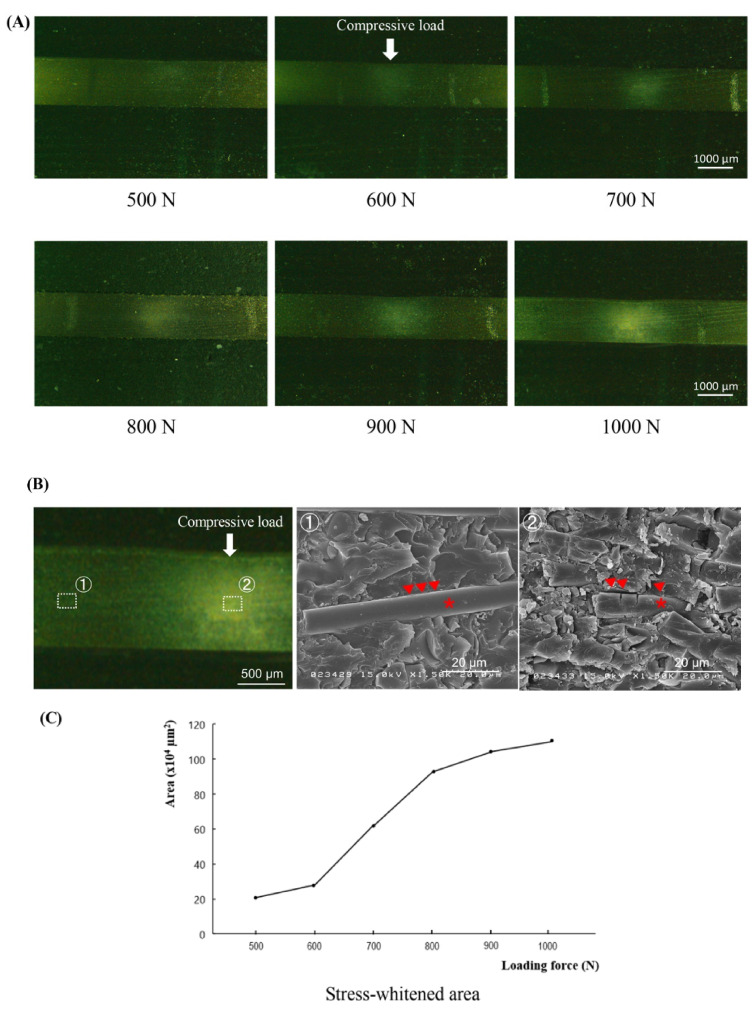
Results of the indentation test following the loading of forces from 500 to 1000 N. (**A**) Light microscopic images of the tested bone pieces in cross-sectional view. Bar = 1000 µm. (**B**) SEM images of the synthetic bone after the loading of force, compared between the intact (①) and stress-whitened (②) areas. Bar = 20 µm. The asterisk represents the glass fiber, and the arrowhead indicates the border between the resin matrix and the glass fiber. (**C**) A line graph shows the mean values of area under stress for each level of force applied.

**Figure 4 materials-14-01825-f004:**
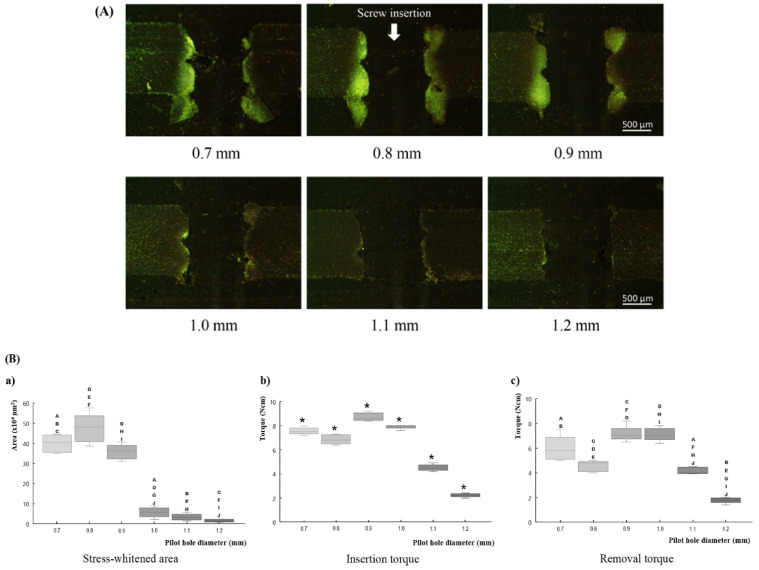
Results of miniscrew insertion test. (**A**) light microscopic images of the tested bone pieces in cross-sectional view (Bar = 500 µm), (**B**) box-plot graphs showing median values. (**a**) stress-whitened area, (**b**) maximum insertion torque, and (**c**) maximum removal torque. The asterisk represents significant differences among all groups (*p* < 0.05). A–J in graph (**a**,**c**) indicate significant differences (*p* < 0.05) between respective pairs of plots.

**Table 1 materials-14-01825-t001:** Physical and mechanical properties of the synthetic cortical bone.

Density (g/mL)	1.7
Ultimate tensile strength (MPa)	90.0
Modulus of elasticity (GPa)	12.4
Compressive yield strength (MPa)	120.0
Compressive modulus (GPa)	7.6

## Data Availability

Not applicable.

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
