# Peer review of "Evaluation of Cortical Bone Microdamage and Primary Stability of Orthodontic Miniscrew Using a Human Bone Analogue"

_materials, 2021, doi:10.3390/ma14081825_

Round 1

Reviewer 1 Report

The manuscript by Chutimont Teekavanich et al. entitled “Evaluation of cortical bone microdamage and primary stability of orthodontic miniscrew using a human bone analogue” deals with the testing of miniscrews in orthodontics. They utilize a human bone analogue to evaluate cortical bone microdamage around the miniscrew and to evaluate the primary stability of the miniscrew to estimate an optimal pilot hole size that could minimize the risk of screw loosening.

This is a nice piece of work that deserves to be published. Despite the fact authors use simple synthetic models, without using animal testing, reducing animal experimentation is a laudable goal that must be followed whenever possible.

Moreover, authors do not overestimate their results, as can be seen in their discussion of results. They clearly pointed out the limitations of the work as can be read: “The potential limitations of this study come from the differences between living tissues and artificial materials. Plastic deformations in synthetic cortical bones cannot distinguish linear cracks or diffuse damage in natural bones, although each microdamaged bone would be repaired or resorped in different ways. Hence, further investigations through clinical experience are required to determine the optimum insertion technique for orthodontic miniscrews”. This is the type of honest statements that is often lacking many studies dealing with biomedical devices.

In order to illustrate the readers please provide an image of the miniscrews used in the present work.

Author Response

Point 1: In order to illustrate the readers please provide an image of the miniscrews used in the present work.

Response 1: We appreciate reviewer’s appropriate comment. We added ‘(Figure 1)’ to line 64 and an image of the miniscrew as Figure 1 to line 65.

Reviewer 2 Report

The authors evaluated the cortical bone microdamage and primary stability of orthodontic miniscrew using a human bone analogue and demonstrated that the estimated pilot hole diameter that could minimize the risk of screw loosening was 1.0 mm for a 1.3 mm diameter miniscrew. The novelty of this paper is mediocre, but the work is solid and might be beneficial to the community. Besides, the reviewer suggested the authors do more careful proof-reading to minimize the grammar errors.

Author Response

Point 1: The reviewer suggested the authors do more careful proof-reading to minimize the grammar errors.

Response 1: We appreciate reviewer’s appropriate comment. We did proof-reading as careful as possible and found some mistakes of misspelling and grammar errors as follow;

Line 156: "fibre" must be changed to "fiber"

Line 178: "the inserton torque. (Figure 3B, c)." must be changed to "the insertion torque (Figure 3B, c)."

Line 286: "repaired or resorped" must be changed to "repaired or resorbed"

Reviewer 3 Report

The presented work is very interesting; the research method and the results obtained were correctly exposed. I appreciated the willingness to use synthetic substitutes without having to exploit animal models for research purposes.
I recommend inserting a reference to the sentence in line 34-35. I also suggest, within the discussion, a greater comparison with similar data already present in the literature which have only been briefly mentioned.

Author Response

Point 1: I recommend inserting a reference to the sentence in line 34-35.

Response 1: In order to support our manuscript, we added a new reference to the sentence in line 34-35.

Nguyen, M,V.; Codrington, J.; Fletcher, L.; Dreyer, C.W.; Sampson, W.J. The influence of miniscrew insertion torque. Eur J Orthod. 2018, 40, 37-44.

Point 2: I also suggest, within the discussion, a greater comparison with similar data already present in the literature which have only been briefly mentioned.

Response 2: We appreciate reviewer’s appropriate comment. As reviewer pointed out, we also think that the discussion will become clearer by adding a comparison with similar data already present in the literature. But unfortunately, as far as we could found, there is only one previous study on the optimum pilot hole diameter for orthodontic miniscrews in Reference 11. Furthermore, regarding the evaluation of microdamage, most of the studies used animal bones, and we could not find any studies that evaluated by using human bone analogue.

Reviewer 4 Report

Thanks for submitting this manuscript, which is evaluated cortical bone microdamage and primary stability of orthodontic miniscrew.

I have carefully read your manuscript with great interest.

I think that it should sound very interesting for readers and this paper overall well written.

This study is well designed and conducted.

I have a few minor comments.

In Figure 2(B), Authors need to change SEM images further high resolution. I cannot distinguish the asterisk.

In Figure 3(B), Author need to clear the meaning of letters (A-J).

What is meaning of A – J, respectively?

Line204: [Same letters (A-J) in each graph indicate significant differences between respective pairs of plots (p > 0.05)]

  • Author more detail to explain the significant differences for readers. Because meaning of significant difference indicate p < 0.05. why mismatched??

Author Response

Point 1: In Figure 2(B), Authors need to change SEM images further high resolution. I cannot distinguish the asterisk.

Response 1: To make Figure 2(B) more clear, we replaced with high resolution images and increased the asterisk.

Point 2: What is meaning of A – J, respectively?

Response 2: As the meaning of A – J is clear, we changed line 203-204 to the following sentence;

‘A - J in graph a) and c) indicate significant differences (p < 0.05) between respective pairs of plots.’

Point 3: Line204: [Same letters (A-J) in each graph indicate significant differences between respective pairs of plots (p > 0.05)]

Author more detail to explain the significant differences for readers. Because meaning of significant difference indicate p < 0.05. why mismatched??

Response 3: We apologized for our mistake in line 204 as reviewer pointed out. We changed ‘p > 0.05’ to ‘p < 0.05’.
